# Disentangling Brillouin’s Negentropy Law of Information and Landauer’s Law on Data Erasure

**DOI:** 10.3390/e28010037

**Published:** 2025-12-27

**Authors:** Didier Lairez

**Affiliations:** Laboratoire des Solides Irradiés, École Polytechnique, CEA, CNRS, IPP, 91128 Palaiseau, France; didier.lairez@polytechnique.edu

**Keywords:** thermodynamics, statistical mechanics, information theory, Brillouin negentropy, Landauer principle, observer

## Abstract

The link between information and energy introduces the observer and their knowledge into the understanding of a fundamental quantity in physics. Two approaches compete to account for this link—Brillouin’s negentropy law of information and Landauer’s law on data erasure—which are often confused. The first, based on Clausius’ inequality and Shannon’s mathematical results, is very robust, whereas the second, based on the simple idea that information requires a material embodiment (data bits), is now perceived as more physical and therefore prevails. In this paper, we show that Landauer’s idea results from a confusion between information (a global emergent concept) and data (a local material object). This confusion leads to many inconsistencies and is incompatible with thermodynamics and information theory. The reason it prevails is interpreted as being due to a frequent tendency of materialism towards reductionism, neglecting emergence and seeking to eliminate the role of the observer. A paradoxical trend, considering that it is often accompanied by the materialist idea that all scientific knowledge, nevertheless, originates from observation. Information and entropy are actually emergent quantities introduced in the theory by convention.

## 1. Introduction

The definition of something is a statement that allows us to recognize it when we see it. For example, it can be a statement of all the characteristics of that thing. Or, it can be the statement that this thing is the name given to a category of elements already defined. Of course, this supposes that the list of all elements in this category is known; otherwise, we would be unable to reliably recognize something as belonging to this category.

For energy, there is no such definition, neither of the first type nor of the second. This is why “*in physics today, we have no knowledge of what energy is*” (R. Feynmann [1]). And this is not a joke; Feynman develops the idea over more than one page.

As a working definition, one can say that “*in the everyday world outside of the formal language of physics, energy is the ability to do anything… Insofar as “does” means “produces a change in what was before”, energy is philosophically the determinant of all observable change*” (E. Hecht) [2]). With this qualitative definition, there is no doubt that knowledge and information are a priori forms of energy. An example is provided by Maxwell [3]: an intelligent being (a demon) produces mechanical work from the information they obtain about a device. If energy cannot come from nowhere, according to the principle of conservation, we see that, a posteriori as well, information is energy.

How to be quantitative about information and its conversion into units of energy? The solution was provided by Shannon’s information theory [4], which, through the intermediary of Gibbs statistical mechanics [5], allowed us to identify the entropy of a system as the uncertainty we have about it. Hence, Brillouin’s law of information [6,7]: decreasing uncertainty by acquiring information has a minimum energy cost equal to that of the corresponding decrease in entropy.

Is that enough? No, according to Landauer. Energy is a physical quantity; information must be too [8,9,10]. However Landauer means by physical something that is not abstract but tangible and material—in other words, a hardware set of data bits. Being materialized, information (understood as data bits) must be erased prior to acquisition. Hence, Landauer’s idea: the energy cost of acquiring information is paid through this preliminary erasure step. This is Landauer’s law on data erasure.

Here, we will see that what may appear as an addition or improvement to the triplet (thermodynamics–statistical mechanics–information theory) is, in reality, totally incompatible with it. The abstract concept of information cannot be confused with its material support (data bit) without introducing many inconsistencies into this theory.

The article is organized as follows. The first part provides a brief overview of this threefold theory (thermostatistics), which in fact forms a whole. Particular attention will be paid to the role assigned to the observer and to conventions. Next, the link between information and energy as established by the demonic engines is presented, together with Brillouin’s and Landauer’s views on how they work. Both are often confused (e.g., [11,12]); thus, this part aims, in particular, to clarify their differences.

Energy is a concept related to the physical world, while information refers to us as intelligent beings endowed with a mind. Even materialists (like myself) have to force themselves to connect the two. That is why, in my opinion, it is an illusion to think that the different conceptions of what science is, of what a theory is, play no role in this problem. Even though physicists often wish to distance themselves from philosophy, an epistemological approach is necessary to understand the interplay between information, energy, probabilities, and the role of the observer. Epistemological positions are always present in the literature on this subject, and in particular in that of Landauer. In most cases, they are hidden or implicit. This does not favor clarity.

We also adopt an epistemological position throughout this article; however, it will be explicit from the outset. It will be that of neo-positivists [13,14,15,16] and those who influenced them [17,18,19,20]. In brief, (1) only two sources of knowledge are admitted: logic and observation; (2) an explanation is a deduction of observations (past and future) from theory; (3) the theory includes inductive laws (generalization of observations), but it must necessarily also include conventions chosen according to their convenience and to the economy of thought they allow. The last part of the paper shows how this position allows us to analyze Landauer’s reductionist idea (beyond its inconsistencies) and how to include emergentism into materialism.

## 2. Thermostatistics

### 2.1. Thermodynamics: Principle Versus Law

Energy is one of the few concepts in physics with universal scope. As such, every scientist knows the fundamental statements on which the theory of thermodynamics, the science of energy transformations, is based. In short:

**Definition**: 
*The state of equilibrium (or state, or equilibrium) of a system is the stationary situation where no change occurs.*


**First principle**:
*The total quantity of energy of an isolated system (the universe) is conserved during any change.*


**Second law**:
*There exists a state quantity, which we call entropy S, such that for any change from state A to state B, $\Delta S_{\mathrm{AB}}=\left(S_{B}-S_{A}\right)\geq \int \;dQ/T$ (Clausius inequality), where Q and T are the heat exchanged and the temperature, respectively.*


But what people are less aware of is the fundamental difference in nature between the last two statements; hence, the terms “principle” and “law” are used to highlight this point (this denomination follows that of Poincaré [18]). A principle is a convention, a law is inferred from induction. Let us clarify this point, starting with the second law.

Clausius inequality expresses that, for a given change of state along a differentiable path, the heat (in temperature units) received by the system from the environment never exceeds a certain limit that depends solely on the initial and final states. This limit is the difference in entropy. It is usually approached as the rate of change along the trajectory slows down in a quasi-static manner. The paths that allow this are called reversible because they can be followed in both directions. The others, especially those that are not differentiable, are called irreversible. In the Clausius inequality, the sign of *Q* refers to the system (positive when received). For simplicity, consider a system with constant internal energy, typically a set of independent entities, such as an ideal gas or a set of bits at a given temperature. Then, for any change, transfers of mechanical work (*W*) and heat (*Q*) with the environment compensate each other (dQ=−dW), so that the Clausius inequality rewrites by considering the reverse change from B to A: $\Delta S_{\mathrm{BA}}\leq \int \;dW/T$. It follows that the entropy difference essentially corresponds to the observed minimum work (in temperature units) required for the system to return to its initial state by a quasi-static path (hence the word “entropy” from Greek entropia “a turning toward”). It is clear that mechanical work and heat can be measured in as many different circumstances as one can imagine. The Clausius inequality results from the generalization of these specific observations, the regularity of which has never been contradicted in two centuries of experiments in this area. The case of equality allows us to define the word “entropy”, but this is just a nominal definition; the core of the second law lies in the inequality. The second law follows from induction.

As for the first principle, it states that the energy of the universe is conserved regardless of the change undergone by a system. In other words, the energy balance is always zero. One could imagine that the first principle also derives from precise measurements of the energy balance of various changes, which would lead to the conclusion that it is apparently always zero, thus allowing these observations to be generalized and elevated to the rank of a universal law. Of course, this assumes that we are able to recognize (identify) energy, whatever form it takes, when we encounter it. However, this is not how it happened, because there is absolutely nothing that allows us to recognize something a priori as a form of energy before it has been added to the list of known forms. And this is precisely this kind of measurement of energy balance that allows for such an addition. The first principle does not follow from inductive reasoning. Rather, it is a hidden—but very incomplete—definition of what energy is. It is incomplete because if a quantity is conserved, that is, constant over time, any function of this quantity is also conserved and therefore constant over time. Therefore, if a quantity is conserved, many others are also conserved. Among them, what should we understand by “energy”? Despite its incomplete nature, we must be content with this definition because there is no other [1].

Although this would be extremely unlikely for the second law, there is no conceptual impediment to discovering a counterexample that would invalidate the generality of any inductive law. However, for a definition such as the first principle, such an invalidation is conceptually meaningless. A definition cannot be invalidated by an observation: “*[A] principle… is no longer subject to the test of experiment. It is not true or false, it is convenient*” [19]. In case an experiment would lead to a non-zero energy balance, we would have the following two options: (1) either we decide that the first principle is no longer appropriate; or (2) we acknowledge that something has escaped our notice and that we have just discovered a new form of energy. Clearly, the second option is more convenient, and that is how all forms of energy were added to the list of those known [14]; for instance, kinetic energy (energy of motion) and rest mass (energy at rest). Ultimately, energy is defined only as a category comprising an undetermined number of items—a number that depends on the state of the art. Energy is an abstract, conventional concept; it is like a universal in the metaphysical sense attributed to this word. However, this universal has a particularity: it includes a number of elements that depend on our knowledge. Therefore, internal energy cannot be considered as intrinsic to the state of a system, nor as intrinsic to certain phenomena. That is to say, energy cannot be considered independent of the observer. “*Energy is […] the determinant of all observable change*” (E. Hecht [2]).

What about entropy? The notion of change is central to the first principle, which defines energy—energy is conserved when everything else changes—but also to the second law, which defines entropy. The latter is not defined in an absolute way by Clausius’ inequality, but only in a relative way, through the measurement of the heat exchanged when the system undergoes a change of state. No observable change means no difference in entropy. We could say the same for any state quantity, for example, temperature. But for temperature, we have means to measure its value for a given state without the need for any change. For entropy, there is no such mean. Entropy is a state quantity that can only be deduced from the observation of a change of state. Ultimately, entropy depends on the ability of the observer to observe these changes, that is, on the observer’s ability to perceive differences. No work is needed to return to the initial state if no change is observed: “*The idea of dissipation of energy depends on the extent of our knowledge*” (J.C. Maxwell [21]).

Like internal energy, entropy also depends on the observer’s knowledge and cannot be considered an intrinsic property of the state of a system.

### 2.2. Statistical Mechanics: Reductionism Versus Emergentism

Statistical mechanics, “*the rational foundation of thermodynamics*” [5], aims to remedy the subjectivity introduced by the observer in thermodynamics. The goal is to calculate everything from Newton’s mechanics of atoms, that is, from the behavior of objects totally independent of us. It is a reductionist approach.

At the atomic scale, everything is in motion. Microscopic configurations (also named phases or microstates) are constantly changing. However, they are so numerous that it is impossible to hope to know that of a system other than in terms of probabilities. Thus, statistical mechanics faces the following two main problems:**Prior distribution**: What phase probability distribution should be used as a starting point for calculations, considering that it cannot be measured?**Equilibrium**: Since phases are never stationary, a definition of equilibrium other than that of thermodynamics is necessary. Which one?

The first problem is generally solved by the ergodic hypothesis, which essentially considers that the phase distribution of snapshots of identical systems (which differ only in their phase) is equal to that of a single system evolving in time. The trajectory of the system in the phase space Γ is supposed to be volume preserving, like that of a deterministic mechanical system (Liouville’s theorem). For two successive points *a* and *b* of the trajectory, if the system is deterministic, the probability for the system to be in *b* knowing it was in *a* is equal to 1. Thus, the probability of *b* is equal to that of *a*, and this holds throughout the trajectory. This results in a uniform probability distribution of phases for isolated systems, thus opening the door to additional calculations and great results. In particular, the Boltzmann distribution for a closed system and the temperature dependence of the partition function [22], which—by identification with known thermodynamics equations—ultimately lead to the famous Gibbs’ result for entropy:(1)$$S=\sum_{i\in \Gamma }p_{i}\ln1/p_{i}$$
where $p_{i}$ is the probability of phase *i*. In the case where the distribution is uniform with $p_{i}=1/W$, one has(2)S=lnW
Clausius entropy, initially defined in thermodynamics solely by its measurable variations, was equal to the minimum amount of heat received by the environment. Thanks to statistical mechanics, it is now understood in absolute terms as the logarithm of the number of possibilities for the microscopic configuration. If understanding means creating connections, then progress is substantial.

However, removing the observer is not without introducing new problems. Where has the observer’s knowledge gone, the information he possesses? The best illustration of this problem is provided by the following two Gibbs paradoxes [23,24]:Joining two identical volumes of the same gas increases the volume accessible to each particle and therefore the total number of possibilities for the system and its Gibbs entropy. However, this occurs without heat exchange and thus without variation of Clausius entropy. The system can return to the initial state at no work, simply by replacing the partition between the two volumes.Mixing two volumes of gas requires work to return to the initial state only if these two gases were initially identified as different. However for statistical mechanics, both cases increase entropy because replacing the partition between the two volumes is not enough to ensure that each particle returns to its original compartment.

Actually, in these two problems, statistical mechanics considers the overall information needed to describe the system (which particle in which compartment), whereas thermodynamics considers incomplete information. “*It is to states of systems thus incompletely defined that the problems of thermodynamics relate*” (J.W. Gibbs [25]).

In thermodynamics, the perception of a change depends not only on the observer’s knowledge, but also on what they consider relevant. In a certain sense, it is true that “*no man ever steps in the same river twice*” (Heraclitus). On the other hand, what makes the identity of the river and ours is not that of the molecules that constitute us: tomorrow, it will still be the Seine that flows through Paris, and I hope to still be myself. The identity of molecules is information that exists, at least for large traceable molecules, but it is considered in thermodynamics by the observer as meaningless or irrelevant in certain cases (e.g., for open systems). The level of information considered relevant is a convention. And it is only in this way, by reintroducing the observer and the information they consider relevant, that Gibbs paradoxes can be resolved for large traceable molecules (see [26] (pp. 13–14) and [27]).

The second problem of statistical mechanics is related to the definition of equilibrium. The current phase is constantly changing. We could simply define the equilibrium as the macroscopic state whose properties take the average values of those of the microscopic configurations. However, this is not sufficient; something is missing. From experience, I know that the equilibrium state of a gas in a room is that in which it uniformly occupies the entire volume and not just a corner. Fluctuations occur that take the system away from equilibrium, but there is something that restores it. Consider a gas inside a cylinder below a piston. We can push or pull the piston and experience a force (Figure 1). The system behaves as a metal spring would. The difference is that for the metal spring, the macroscopic equilibrium reflects the microscopic equilibrium of the atoms that are in a potential well. Such a potential well does not exist for the atoms of an ideal gas. One could argue that the equilibrium position of the piston corresponds to the equality of internal and external pressures, and calculate the pressure as being related to the number of atomic collisions on both sides. In this way, as for the metal spring, macroscopic forces would ultimately be explained by microscopic ones. However, the calculation inevitably starts with the assumption that the atom density near the surface is equal to that of the gas at equilibrium for a given piston position. In other words, the calculation starts with the very assumption it intends to prove. This is circular reasoning.

In reality, the force exerted on the piston is an emergent property, just like pressure and entropy. It cannot be derived consistently from the microscopic scale and the laws we already have at our disposal. An additional ingredient is necessary, which must be postulated upstream in the theory through the conventional definition we give of macroscopic equilibrium. Just as a metal spring minimizes its potential energy at equilibrium, it would be convenient if a volume of gas optimized something. In thermodynamics, we already have the second law that says that dS≥0 for an isolated system. If we postulate that entropy is maximum at equilibrium, the problem is resolved [28], and the restoring force we are seeking naturally emerges. The equilibrium would become an attractor.

This is where the problem arises with such a postulate within the framework of statistical mechanics as we have presented it so far. According to Poincaré’s recurrence theorem [29], a volume-preserving dynamical system is recurrent and admits no attractor. This property is inherent to the ergodic hypothesis and to its conception of probabilities as frequencies of occurrence: frequency implies recurrence. The solution would consist in abandoning the ergodic hypothesis. However, then comes back the problem of the prior probability distribution of phases. It is a vicious circle. Finally, the best approach would be to admit that “*When one does not know anything, the answer is simple. One is satisfied with enumerating the possible events and assigning equal probabilities to them.*” (R. Balian [30]). This is known as the fundamental postulate of statistical mechanics, but in reality, nothing more than Laplace’s “principle of insufficient reason” [31]. The problem is that, as it stands, it is unfounded. It is a synthetic a priori knowledge: “*It cannot be that because we are ignorant of the matter we know something about it*” (R.L. Ellis [32]).

Information theory provides us precisely with the missing pieces that we lack. It reintroduces the observer’s knowledge (because information is meaningless without an observer) and solves the problem of prior probabilities by making the principle of insufficient reason analytical.

### 2.3. Information Theory: The Return of the Observer

Shannon [4] tackled a problem that initially appeared very different from that of thermostatistics, but which ultimately turned out not to be so distant: the lossless compression of a message. Shannon began by abstracting away the meaning of the message and by modeling its emitter as a source of a random variable taking values in a set Γ of possible characters. In this framework, he mathematically demonstrated the following two results:In no case, the average number of bits per character is less than(3)$$H=\sum_{i\in \Gamma }p_{i}\log_{2}\left(1/p_{i}\right)$$*H* is named quantity of information emitted by the source and by identification with Equation (Equation 1), S=H×ln2 is its entropy.Within a factor, *H* (and thus *S*) is the only measure of uncertainty on the upcoming character that is (1) continuous in *p*; (2) increasing in W=1/p for uniform distributions; and (3) additive over different independent sources of uncertainty.

These results are general and apply regardless of the random variable, and in particular to the constantly evolving phases of any physical dynamic system. It follows that entropy, having first been defined as a quantity of dissipated heat (Clausius) and then as the logarithm of a number of possibilities (Gibbs), is now—according to Shannon—the uncertainty that the observer has concerning the phase of the system. Here again, its understanding had made great progress.

These two results were quickly recognized as a major advance in thermostatics. Brillouin, with his law of information [6,7] and Jaynes, with the maximum entropy principle [33,34], were the first. The first point will be presented in the following section; therefore, we provide a brief overview of the second here.

How to describe or mentally represent, in a rational way, something that we only partially know? That is, in reality, the central question of science. The description must account for all pieces of information; otherwise, it would be incomplete, but it must not invent information that comes out of nowhere (which would constitute a synthetic a priori knowledge). For instance, this is the problem of linear regression: for the solution (the description) to be unique, the degree of the polynomial must be less than the number of points. Solving such a problem amounts to seeking a unique description that maximizes the uncertainty about the information we do not have. For a description involving a probability distribution, Shore and Johnson [35] showed that maximizing its entropy (instead of another possible measure of uncertainty) is the only procedure leading to a unique solution.

Hence, the theorem of maximum entropy: the best prior probability distribution that accounts for our knowledge is that of maximum entropy. For instance, Shannon [4] showed that “*when one does not know anything…*” (Balian) the maximum of entropy is obtained for a uniform distribution. This provides a mathematical basis for Laplace’s principle of insufficient reason and makes it analytical.

By abandoning the ergodic hypothesis, there is no longer any obstacle or inconsistency in postulating by convention that equilibrium is an attractor—that is, in defining equilibrium as the state of maximum entropy of the phase distribution (which is also the state of maximum entropy of variables whose distributions are similarity-invariant in form [36], such as the density). This is the principle of maximum entropy [33,34]. This principle, like any other definition, is conventional. It is not required to be checked by experiment. And, thankfully so, because it would not be possible. This principle is just convenient. Without introducing inconsistency, this allows us to establish a deductive link between theoretical statements (the definition of equilibrium plus the second law) and experiments (the observed restoring force towards equilibrium), that is, to explain and account for the latter.

## 3. Information and Demons

### 3.1. From Maxwell to Szilard

The connection between information and energy was established by Maxwell [3]. He imagines an intelligent being (say, a demon or a computer) capable of tracking particles and, from this knowledge, extracting energy from the system, which would otherwise be impossible. A simpler version of such an engine is that of Szilard [37], shown in Figure 2. Applying the first principle of thermodynamics, which states that energy cannot, by definition, come from nowhere, forces us either to admit that information is a form of energy or to formulate another definition of energy. Undeniably, the first alternative is more convenient.

However, it is necessary to be more quantitative and explicit. This is precisely the purpose of Brillouin’s law of information and also that of Landauer’s about data erasure.

### 3.2. Brillouin’s Negentropy Law of Information

Brillouin’s reasoning consists of three premises:**Clausius**: The negative of the entropy difference ΔS experienced by a system (at a given *T*) is the minimum work *W* that must be performed on the system for the change to take place: W≥−TΔS.**Gibbs**: Entropy *S* is related to the probability distribution of possible microstates.**Shannon**: The Gibbs formula for *S* is actually to a factor of ln2 that of the uncertainty *H* on the actual microstate: S=Hln2
And one deduction:
4.**Brillouin**: Therefore, reducing uncertainty by acquiring information requires minimum work: W≥−TΔHln2. For one bit of acquired information ΔH=−1; thus(4)$$W_{\mathrm{acq}/\mathrm{bit}}\geq T\ln2$$
where $W_{\mathrm{acq}/\mathrm{bit}}$ is the minimum work that we have to provide (and that will be dissipated as heat) per bit of acquired information. This is the Brillouin’s negentropy law of information [6,7], called “principle” by Brillouin [6] and sometimes referred to as Szilard’s principle [38], since Equation (Equation 4) can be derived from the operation of the Szilard engine. Here, however, it is referred to as a “law” for the sake of consistency with Section 2.1, because it is derived from the second law. With Brillouin’s equation, the energy balance of the Szilard engine is zero, as required.

Let us consider a body of mass *m*, lifted to a height Δh. Classical mechanics tells us that the increase in potential energy is mgΔh. This result does not impose any particular steps for the change in height and does not specify where the mechanical work is performed. However, the result concerning the net change in potential energy is general and valid in all cases, regardless of the path taken.

Brillouin’s law works in exactly the same way. Entropy, and consequently the uncertainty about a system or the information we lack to describe it, is a state quantity whose variation is independent of the path connecting two states. Thus, Brillouin’s law does not tell us where the energy cost of acquisition is paid in the operation. And this is inherent to the definition of what a state quantity is. Acquisition may not be a simple operation, but may consist of other, more elementary operations, which may differ depending on the specific data acquired. However, in the end, regardless of how the acquisition is performed, it will cost at least Tln2 on average per bit.

Brillouin’s law is a syllogism and states nothing more than its premises—in this case two mathematical results and the second law. It follows that its generality is as robust as that of the second law.

### 3.3. Landauer’s Law on Data Erasure

Landauer’s competing argument [8] is based on two main ideas. The first is that “*information is not a disembodied abstract entity; it is always tied to a physical representation. It is represented by engraving on a stone tablet, a spin, a charge, a hole in a punched card, a mark on paper, or some other equivalent*” [10]. In other words, information is always supported by hardware with at least two discernible different configurations (two values: 0 or 1), namely a data bit. The second idea is that the acquisition of one bit of information necessarily passes by the erasure of its supporting data bit. The reasoning is as follows:Any intelligent being has a finite memory; thus, the infinite cyclic acquisition of information about a dynamical system necessarily requires the erasure of data bits.Erasing a data bit (a thermodynamical system) consists of setting it to an arbitrary value (say 0). The procedure must be able to work for a known or unknown initial value (i.e., it must be the same in both cases). This constraint automatically implies a two-step erasure process (see Figure 3):
(a)Free expansion of the phase space by a factor 2, leading the system to an undetermined standard state (state S).(b)Quasi-static compression of the phase space by the same factor leading the system from state S to state 0.The first step does not involve any exchanges with the environment, whereas the second dissipates at least Tln2 of heat, or equivalently, at constant internal energy (i.e., at constant temperature for a set of independent data bits), it requires a minimum work. The net balance of the two yields(5)$$W_{\mathrm{erase}/\mathrm{bit}}\geq T\ln2$$
which leads, as Brillouin’s law does (Equation (Equation 4)), to a zero energy balance for Szilard engine. The difference lies in the fact that, according to Landauer, erasure is a necessary step, and it is at this very point that the energy cost of data acquisition is paid.Landauer’s law can be examined from two different angles:
Restricted context of data erasure: Does erasing one data bit really require at least Tln2 (Landauer’s limit) of work?To my knowledge, all the authors (see, e.g., [39,40,41,42,43,44]) who have addressed this question actually used Landauer’s procedure that consists of (a) free expansion; (b) reversible compression. In this way, the authors simply test the second law rather than the novel aspect of Landauer’s idea; therefore, their results are unsurprising, as there is no error in Landauer’s calculation. What is new in Landauer’s assertion? It is the claim that there is no alternative erasing procedure. Thus, the only way to confirm Landauer’s law would be to search, in vain, for an alternative. However, this is not what was done. In reality, even within the restricted context of data erasure, several points of Landauer’s reasoning are questionable: the imperative to begin erasure with expansion, and the necessity of this expansion to be thermodynamically irreversible. Counterexamples have been provided [45,46,47], which invalidate the generality of Landauer’s procedure.Broader context of information acquisition and loss: Does data erasure equate to information loss? Can Landauer’s law on data erasure be considered as the missing link between information and energy, something that would replace Brillouin’s law of information? In the following section, we focus on the inconsistencies this idea introduces by confusing information and data.

## 4. Information Versus Data

The confusion between information and data also corresponds to the confusion between the loss (or deletion) of a bit of information and the erasure of a bit of data. The first refers to the observer’s knowledge, whereas the second only involves a physical object independent of them. A similar confusion in statistical mechanics has already led to inconsistencies. It is, therefore, not surprising to see others with Landauer’s idea. These inconsistencies make Landauer’s law incompatible with that of Brillouin—that is, incompatible with the triplet (thermodynamics–statistical mechanics–information theory).

### 4.1. Total Versus Incomplete Information

In thermodynamics, setting a data bit into the S state by removing the partition between two compartments containing a single particle (first step of Landauer erasure) is reversible or not, depending on whether the initial value of the bit is unknown or not [48] (see Figure 4). This problem is the same as that of the Gibbs paradox, and the solution is also the same: thermodynamics considers the case where the observer has incomplete information about the system, whereas Landauer, in his reasoning, implicitly assumes that the observer has all information at his disposal. There is no solution to this paradox other than those that reintroduce the observer, because in reality, taking these two cases into account (as Bennett does [49], a defender of Landauer’s thesis), already amounts to taking the observer into account.

In addition, the constraint of a single path for erasure, regardless of the initial value of the bit, stems from the requirement to work with unknown values. But this case is thermodynamically reversible. For known values, there is nothing preventing the use of two different reversible paths. The use of a single erasing procedure for known and unknown cases is an arbitrary choice by the observer. Here again lies a clear inconsistency between the intention of the theory and its result—the desire to eliminate the observer while reintroducing him.

### 4.2. Global Concept Versus Local Object

Information is a concept that must not be confused with its material embodiment. The proof is provided by Landauer himself: “*Information […] is represented by engraving on a stone tablet, a spin, a charge, a hole in a punched card, a mark on paper, or some other equivalent*” [10]. The material embodiment may change, but information remains the same. The denial of this difference leads to severe inconsistencies.

A fundamental property of information is that it cannot be given twice. We can duplicate bits of data encoding certain information so that the storage space occupied is twice as large as before, but the corresponding amount of information remains unchanged. Conversely, if multiple copies of the same data bit are erased, the corresponding information will only be lost when the last copy is erased. To assert that information has been erased, one must take into account not only what happened locally, but also the existence or not of a copy somewhere in a more global space.

Information is a global concept, whereas a data bit is a local object.

It could be argued that this is playing with words and that information can have different meanings. Actually, the common meaning of information as “*The imparting of knowledge in general. Knowledge communicated concerning some particular fact, subject, or event*” (Oxford Dictionary) is also that adopted by Shannon: if the language used for a message is known, the message can be further compressed, however indicating the language twice does not yield any additional space savings. This meaning is likewise shared by proponents of Landauer’s idea (“*But what is information? A simple, intuitive answer is “what you don’t already know”*” [41]. “*If someone tells you that the earth is spherical, you surely would not learn much*” [43]).

The denial of the difference between information and data and the attempt to materialize information is culminating with the supposed “information-mass equivalence” [12,50,51,52] that leads to paradoxes in relation to the corresponding mass deficit. For instance, in the framework of this equivalence, consider a data bit encoding a bit of information. Duplicate the data bit and make the original and the copy physically independent. Erase one of the two. The quantity of information remains unchanged, so the erased data bit does not display any mass deficit. Erase the second bit. Information is lost that corresponds to a mass deficit. How does the second bit “know” that the first has been erased? This contradicts the hypothesis of independence. But if we suppress this hypothesis and assert that independence is impossible, does that not mean that information is not a property of the local data bit, but a property of a global system?

Yet another variation of this paradox. Consider a data bit encoding a bit of information whose initial value is 0. Erase the data bit (set its value to 0). There is no longer information stored by the data bit. Erase the bit again. What is the difference between the two erasures? If there is no difference, an eventual mass deficit should not be due to any loss of information. If there is a difference, how does the data bit “know” that it has already been erased? Only a global observer could.

A data bit has a physical meaning at all scales—from that of a single particle in a two-compartment box, to the macroscopic scale. But information is meaningful only when it involves an observer; information is meaningful only at the macroscale. Information, even the smallest piece, is an emergent concept.

### 4.3. Pair of States Versus Path

According to Landauer, “*Logical irreversibility is associated with physical [thermodynamical] irreversibility*” [8]. This is a claim of generality of Landauer’s law and is probably the most problematic point.

Logical irreversibility refers to a loss of information (hence Landauer’s idea of establishing a link with data erasure) that we possess about a system. A loss of information is an increase in the quantity of information we lack to describe the system, or an increase in the uncertainty about it; in other words, an increase in its entropy. Logical irreversibility is simply a positive change in entropy. It is solely related to a change in a state quantity. This is a consequence of the Clausius definition of entropy as a state quantity and of the successive mathematical results of Gibbs and Shannon equating quantity of information and entropy, all condensed in Brillouin’s law of information.

Thermodynamic irreversibility, on the other hand, is a property not linked to a difference between two states, but to the path used to connect them.

How does Landauer arrive at the conclusion that the two irreversibilities are associated? In fact, Landauer forces the path to go through a particular non-differentiable sequence of events, so that a property which initially in the theory is related solely to a peculiar path (thermodynamic irreversibility), becomes also a property of an ordered pair of states (logical irreversibility). This viewpoint either empties the very definition of a state of all meaning or transforms information and thus entropy into quantities that are not state quantities. In both cases, Landauer’s idea is incompatible with thermodynamics and information theory and leads to inconsistencies.

Consider a system changing from A to B, with $S_{A}<S_{B}$. The change is logically irreversible (the information we lack increases); thus, according to Landauer, it is also always thermodynamically irreversible. But the change in the other direction from B to A is logically reversible; thus, according to Landauer, it could be thermodynamically reversible. In summary, the change from A to B would be thermodynamically irreversible, and that from B to A would be thermodynamically reversible. What exactly would “thermodynamically reversible” mean in this context?

In fact, Landauer erasure and Brillouin’s law only lead to the same result in the context of a cyclic data acquisition (such as that of a Szilard engine). Outside of this context, they do not. Brillouin’s law concerns the acquisition of information; Landauer’s law concerns the erasure of data; hence, the contradiction if we equate the two. In the context of a Szilard engine and Landauer’s scenario, the data erased before acquisition relates to the engine’s state during the previous cycle. This data is essentially obsolete; it concerns information about the past. The erased data no longer contains information about the engine’s current state.

## 5. Conclusive Epistemological Point

In the literature, Landauer’s law on data erasure is generally presented as a decisive improvement to information theory that provides us with the missing key to understanding the link between information and thermodynamics. For instance, one can read: “*The Landauer principle is one of the cornerstones of the modern theory of information*” (Herrera [12]).

This situation is very strange because there is absolutely no experiment and no observation, which can be explained by Landauer’s law, and that would not be explained without it. On one side, the link between information and energy, as established by the Maxwell or Szilard engine, is perfectly explained by Brillouin’s law. On the other side, the erasure of data bits, whatever their form, is also perfectly explained by thermodynamics. Landauer’s law adds absolutely nothing to the theory, except for inconsistencies that make it incompatible with thermodynamics and information theory. So, how do we conceive its popularity? It is as if an explanation that consists solely of deducing observations (past and future) from a set of theoretical statements (principles and laws) is insufficient. Yet, that is precisely the only thing expected of theory.

The popularity of Landauer’s law can only be due to a misconception of what concepts are in physics. There is no physical theory without concepts. But concepts such as energy and information (but also that of field and many others) are not material entities, although they are fully physical, in the sense that they play a crucial role in current physical theories. They are, in fact, conventions. The meaning of any concept, what it encompasses, and its definition, is a convention because we cannot see it, and consequently, establish a direct link between what we see and what we mean. We cannot show a concept to someone.

“*Information is physical*” (Landauer [9]). “*Information is not a disembodied abstract entity*” (Landauer [10]). Landauer’s law on data erasure is, in reality, an attempt to materialize information, which, without Landauer, would remain an abstract concept. According to his idea, the smallest piece of information is actually localized and materialized under the form of a data bit. In this sense, Landauer’s idea belongs to the reductionist branch of materialism. It is in direct continuity with the initial intention of Gibbs’ statistical mechanics. Here, the purpose is neither to contest materialism nor to deny the utility of reductionism when it allows an economy of thought. It is simply to deny reductionism as a profession of faith or a creed.

The concept of energy is widely used in physics, although it is a pure concept. There is no such thing as an “energy particle” (an object) that could exist independently of us. Energy is like a universal and nobody has a problem with that. Why not consider information in the same way?

Attached to materialism and to “*the scientific conception of the world*” [13] (neo-positivism) is the idea that the only two sources of knowledge are that underlying mathematics, with no other synthetic a priori knowledge, and that of observations. The former imposes a strict requirement of consistency in the statements of the theory, while the latter obliges us to take into account the role of the observer, since there is no observation without an observer.

This is where a potential problem of misunderstanding lies. Accounting for the observer introduces information. A concept that is emergent since the observer is only meaningful at the macroscopic scale. Hence, the danger of falling into a form of holism that would consist of considering emergence as synthetic a priori knowledge. In other words, precisely what neo-positivism intends to reject. This apparent contradiction introduced by emergence into materialism probably explains the search for a reductionist solution like Landauer’s.

Actually, a theoretical statement involving an emergent property, such as the principle of maximum entropy, is not a synthetic a priori knowledge; it is a convention. A convention that is exactly of the same nature as that of the first principle of thermodynamics or Euclidean geometry [16,18]. The choice of one convention or another is guided solely by its convenience and, in particular, by the economy of thought [14] it provides: the simplest is the best. The prior probability distribution of phases of a system, which is directly deduced from the principle of maximum entropy, is in this sense also a convention (it is directly derived from a convention). It does not pretend to reflect the reality of the system. It is not the system that maximizes the probability distribution of its phases, but our conventional representation of the system that must do so if we want to be rational.

Emergentism is generally and historically divided into two categories [53]: ontological and epistemological. The former considers it to be inherent in the essence of things, whereas the latter holds that it is imposed only on us by our necessary limited knowledge. However, with regard to experience, both lead to the same result. The difference is a metaphysical question out of the scope of science. The neo-positivist position [13,16], inspired by Poincaré’s conventionalism [18], which is defended here, makes it possible to avoid this problem and constitutes a third way that could be called “conventional emergentism” and is not so far removed from pragmatism [54,55].

## Figures and Tables

**Figure 1 entropy-28-00037-f001:**
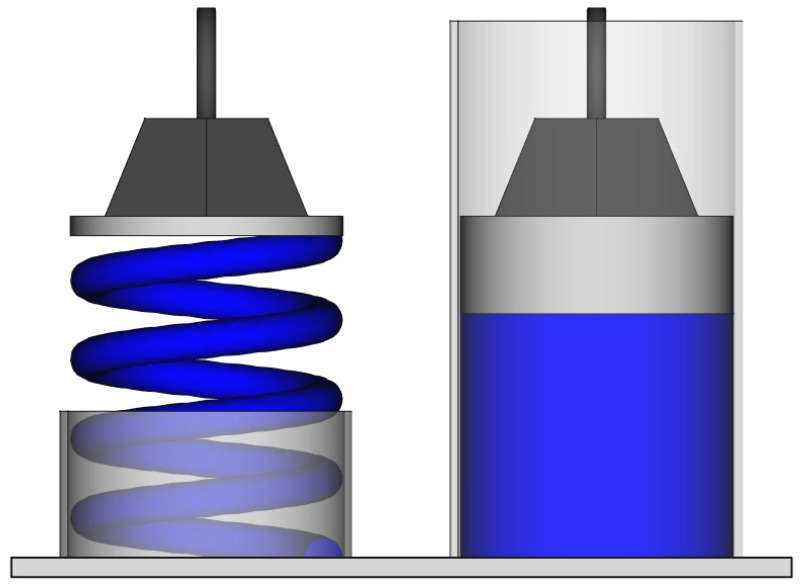
A piston compressing either a metal spring (**left**) or a volume of gas (**right**) experiences a restoring force when it moves away from equilibrium. For the metal spring, this force has a microscopic origin. For the gas, it is emergent.

**Figure 2 entropy-28-00037-f002:**
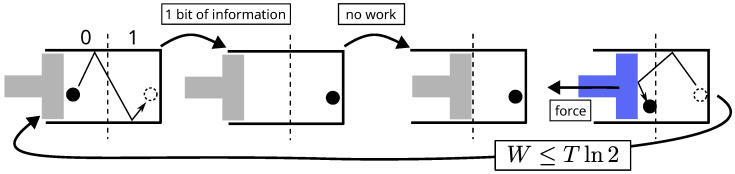
Cycle of the operations of Szilard’s engine. By knowing the location of the particle, a demon (say a computer) can produce mechanical work with no other cost than that of acquiring the information. The first principle implies that information is energy.

**Figure 3 entropy-28-00037-f003:**
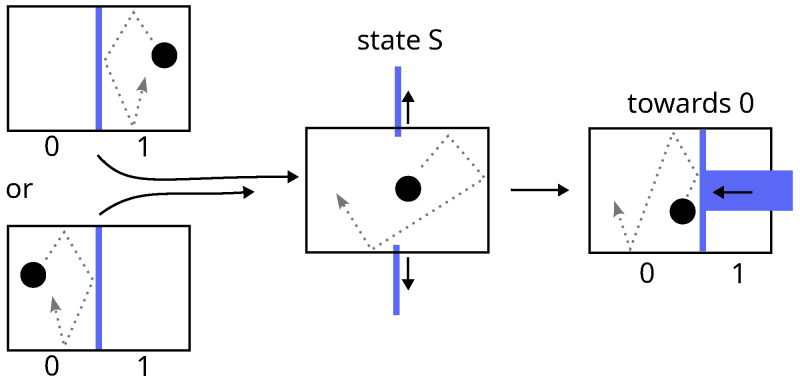
Landauer erasure of a data bit consisting of a single particle in a two-compartment box. The procedure must be independent of the initial position of the particle: neither a simple shift of the particle towards 0 by means of two pistons keeping the phase-space volume constant, nor a quasi-static expansion with a single piston, is possible. According to Landauer, the first step is necessarily a thermodynamically irreversible free expansion.

**Figure 4 entropy-28-00037-f004:**
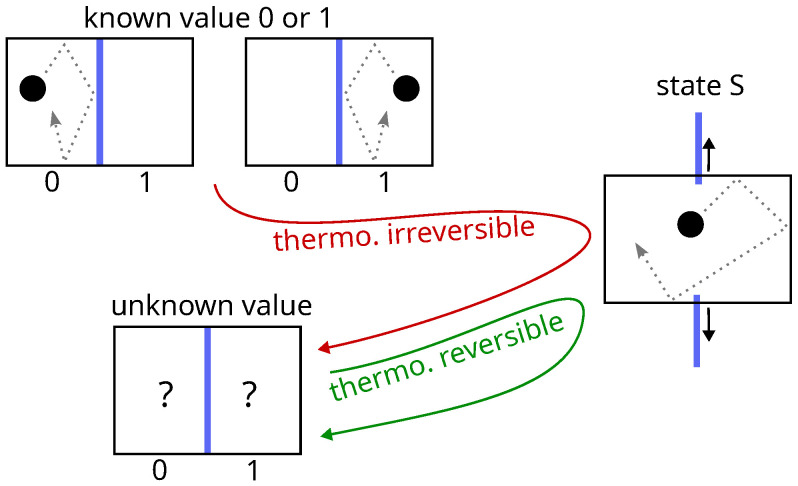
Setting a data bit in state S by merging two compartments is thermodynamically irreversible (red path) if the initial value is known (ΔS=ln2), but it is thermodynamically reversible (green path) if the initial value is unknown (ΔS=0). This is reminiscent of the Gibbs paradox of mixing. Landauer’s viewpoint conflicts with thermodynamics.

## Data Availability

No new data were created or analyzed in this study.

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
