# Peer review of "Disentangling Brillouin’s Negentropy Law of Information and Landauer’s Law on Data Erasure"

_entropy, 2025, doi:10.3390/e28010037_

Round 1

Reviewer 1 Report

Comments and Suggestions for Authors

I do not agree with the author of this paper in most of the important points addressed in the manuscript. In spite of this, I strongly recommend this paper for publication for two reasons:

A. The paper is  think-provoking.

B. The paper touches the deepest foundations of physics and its understanding by humans. The Entropy journal is exactly an appropriate forum for discussion these problems. Remarks:

A. In the text: "The definition of something is a statement that allows us to recognize it when we see it. For example, it can be the statement of all the characteristics of that thing. Or, it can be the statement that this thing is the name given to a category of elements already defined. Of course, this supposes that the list of all elements in this category is known, otherwise we would be unable to recognize every time an element as belonging to this category. For energy, there is no such definition, neither of the first type nor of the second. This is why “in physics today, we have no knowledge of what energy is” (R. Feynmann)."   

I do not agree that: "The definition of something is a statement that allows us to recognize it when we see it". This "definition of definition" does not work neither in mathematics nor in physics. According to the Hilbert definition: mathematical axioms are definitions of mathematical objects, such as points, lines. etc. The same cum grano salis is true for physics. The joke of Richard Feynman should not be taken too seriously. We know, what is energy:

 Energy is a scalar quantity defined from the time-translation symmetry of physical laws and conserved under that symmetry.  Energy is a generator of time evolution and a conserved Noether charge. 

This is a Hilbert-like definition of energy. We define "energy" via physical laws/axioms, i.e. its conservation emerging from the space-time symmetries. 

2. We also know what is information.

Information is the physically realizable distinction between alternative possible states of a system that can be preserved, transmitted, and transformed. Information quantifies how many physically distinguishable states are available to the system.

Equivalently:

Information is the reduction of physical uncertainty about the state of a system, constrained by physical laws. Information is a physical quantity whose manipulation necessarily involves energy, entropy, and heat. This is also Hilbert-styled definition of information. And here the Landauer principle becomes useful. 

3. As to the Landauer principle: the Landauer principle is fruitful and instructive. It is not exact, but fruitful and instructive. No physical law is exact. All of physical laws work in idealized, model systems. Energy conservation occurs only in the isolated, model physical systems which do not exist. However, the energy conservation law is extremely fruitful and instructive. The same is true for the Landauer principle.

4. The Landauer principle becomes much more clear in the context of other limiting physical principles, see:  

Bormashenko, E. Landauer Bound in the Context of Minimal Physical Principles: Meaning, Experimental Verification, Controversies and Perspectives. Entropy 2024, 26, 423,

which are also no exact but fruitful.

Again, I strongly recommend this paper for publication.

Author Response

Comment 1:
I do not agree with the author of this paper ... In spite of this, I strongly recommend this paper for publication for two reasons:
******************************
Response 1:
This is entirely to your credit.
****************************************************

Comment 2:
Remarks:
...
The joke of Richard Feynman should not be taken too seriously. We know, what is energy:
Energy is a scalar quantity defined from the time-translation symmetry of physical laws and conserved under that symmetry...
This is a Hilbert-like definition of energy. We define "energy" via physical laws/axioms, i.e. its conservation emerging from the space-time symmetries.
************************
Response 2
The sentence of Feynmann reported in the introduction cannot be considered a joke for two reasons relative to the form:
1) It is taken from Feynman's lectures intended for students who are supposed to know nothing about this subject and therefore cannot distinguish between a joke and a serious statement.
2) This is not simply a single isolated small talk, but an entire page is devoted to this point in Feynmann's lectures.

Regarding the content, in summary: "Energy is defined by a conservation principle", is according to you a sufficient definition of energy. Actually, it cannot be a true definition, because if a quantity is conserved, meaning constant over time, any function of this quantity is also conserved and constant over time. So that, if a quantity is conserved, many others are also conserved. Among them, what should we understand by "energy"?

The paper has been modified to account for your remark (see lines 120-125)
****************************************************

Comment 3:
2. We also know what is information.
...
************************
Response 3:
Yes, and information is not data. This is my point.
****************************************************
...

Comment 4:
3. As to the Landauer principle: the Landauer principle is fruitful and instructive...
************************
Response 4:
No, I do not agree and this the main point of the paper. 
If by "fruitful" we mean "productive", what should a "fruitful principle" be? It should be a principle that allows us to produce many new explanations. There is absolutely no new explanation produced by Landauer's principle that is not explained otherwise.
****************************************************

Reviewer 2 Report

Comments and Suggestions for Authors

In this paper, the author presents a difference between Brillouin's law and Landauer's law, and claims that the latter contains inconsistencies in reasoning from the outset. 
I think the author is pointing out a point that no one has noticed before. However, 
some papers claim that the Landauer's law has been confirmed and derived in some ways (the author should know them). How would you counter that argument? Adding the flaws in those papers would strengthen your argument. Wikipedia features several such papers that make that claim. Thus, focusing on just a few of them would be sufficient.

- References 23 and 24 require corresponding websites.

This manuscript can be recommended for publication if the author considers the above-mentioned points.

Author Response

Comment 1:
... some papers claim that the Landauer's law has been confirmed and derived in some ways (the author should know them). How would you counter that argument? Adding the flaws in those papers would strengthen your argument.
************************
Response 1:
To account for your remark, the paper has been modified to point out the flaws of these "confirmations", see lines 375-394.

In short, these authors did not delete information they erased data-bits, and moreover they erased data-bits following Landauer's procedure. So that their results is not surprising. Landauer's principle is the claim that 1) erasing data-bit is equivalent losing information (it is not); 2) erasing data-bit can only be achieved by following Landauer's procedure (it is not true, see counterexamples).
****************************************************

Comment 2:
- References 23 and 24 require corresponding websites.
************************
Response 2:
This has been corrected.
****************************************************************************